# Corrosion Products from Metallic Implants Induce ROS and Cell Death in Human Motoneurons *In Vitro*

**DOI:** 10.3390/jfb14080392

**Published:** 2023-07-25

**Authors:** Hannes Glaß, Anika Jonitz-Heincke, Janine Petters, Jan Lukas, Rainer Bader, Andreas Hermann

**Affiliations:** 1Translational Neurodegeneration Section “Albrecht Kossel”, Department of Neurology, University Medical Center Rostock, University of Rostock, 18147 Rostock, Germany; hannes.glass@med.uni-rostock.de (H.G.);; 2Biomechanics and Implant Technology Research Laboratory, Department of Orthopedics, University Medical Center Rostock, University of Rostock, 18057 Rostock, Germany; 3Center for Transdisciplinary Neurosciences Rostock (CTNR), University Medical Center Rostock, University of Rostock, 18147 Rostock, Germany; 4Deutsches Zentrum für Neurodegenerative Erkrankungen (DZNE) Rostock/Greifswald, 18147 Rostock, Germany

**Keywords:** metal implant, cobalt, nickel, chromium, induced pluripotent stem cells, motoneuron, neurodegeneration, neuropathy, reactive oxygen species

## Abstract

Due to advances in surgical procedures and the biocompatibility of materials used in total joint replacement, more and younger patients are undergoing these procedures. Although state-of-the-art joint replacements can last 20 years or longer, wear and corrosion is still a major risk for implant failure, and patients with these implants are exposed for longer to these corrosive products. It is therefore important to investigate the potential effects on the whole organism. Released nanoparticles and ions derived from commonly used metal implants consist, among others, of cobalt, nickel, and chromium. The effect of these metallic products in the process of osteolysis and aseptic implant loosening has already been studied; however, the systemic effect on other cell types, including neurons, remains elusive. To this end, we used human iPSC-derived motoneurons to investigate the effects of metal ions on human neurons. We treated human motoneurons with ion concentrations regularly found in patients, stained them with MitoSOX and propidium iodide, and analyzed them with fluorescence-assisted cell sorting (FACS). We found that upon treatment human motoneurons suffered from the formation of ROS and subsequently died. These effects were most prominent in motoneurons treated with 500 μM of cobalt or nickel, in which we observed significant cell death, whereas chromium showed fewer ROS and no apparent impairment of motoneurons. Our results show that the wear and corrosive products of metal implants at concentrations readily available in peri-implant tissues induced ROS and subsequently cell death in an iPSC-derived motoneuron cell model. We therefore conclude that monitoring of neuronal impairment is important in patients undergoing total joint replacement.

## 1. Introduction

The number of joint replacement procedures is increasing worldwide every year [1]. Due to surgical and material advancements, the outcome for the patients has become better, encouraging even more adults, especially younger ones, to undergo surgery [2,3]. Although state-of-the art joint replacements are expected to last 20 years or longer, they do wear out and need to be replaced [4]. The wear and corrosive products of metal components are the main cause for implant failure due to aseptic loosening [5]. The products released into the peri-implant tissue are metal nanoparticles and cations, which are then distributed systemically [6,7,8]. The most prominent metal alloys used in orthopedic surgeries are titanium–nickel (e.g., nitinol), titanium–aluminum–vanadium (e.g., Ti-6Al-4V), and cobalt–chromium–molybdenum (e.g., Co-28Cr-6Mo) [9,10,11]. Co, Ni, Cr, and Ti are known to cause oxidative stress and DNA damage, which lead to apoptosis and necrosis [6,12]. Systemic concentrations of metal ions in patient blood samples have been intensively studied. Although the concentration in patients with acute poisoning can reach up to 400–640 μg/L for cobalt and 50–80 μg/L for chromium, a concentration of 2–7 μg/L is already considered indicative of excessive wear and can lead to implant loosening [13,14]. Peri-implant concentrations of metal ions can exceed these values even further, up to 397.8 mg/L Co, and show a shift to a highly increased Cr concentration [15]. This is due to the generation of stable CrPO_3_ salts that are retained in the peri-implant tissue [6,7]. Uptake of Ni^2+^ and Co^2+^ is mediated by Ca-, Mg-, or Fe-channels [7]. 

Detrimental effects of these high ion concentrations on bone marrow structure and especially osteoclastogenesis have already been shown [16]. It is known that Co stimulates HIF-1 signaling and is frequently used as chemical hypoxia model [17]. Downstream effects include upregulation of HIF1α but not HIF2α targets, mTORC1 activation, metabolic shift towards glycolytic metabolism, and pro-inflammatory and ER-stress response by activation of IRE1 [18].

Some studies suggest metal endoprosthetic implants as a risk factor for neurodegenerative diseases. Since neurons are highly susceptible to DNA damage and oxidative stress, they might be particularly vulnerable against a chronic systemic increase in metal ions after joint replacement procedures [19]. Thanks to the abovementioned advances in surgical procedures and biocompatibility of materials used in total joint replacement, more and younger patients are undergoing these procedures. This, however, also means that the body is exposed to such metal alloys much longer. Peripheral neuropathy is the prototype clinical phenotype of peripheral neuronal toxicity, and since motoneurons are in direct contact with muscles and reside in relative vicinity to joints, their exposure to these ions might be particularly high compared to central nervous system neurons such as cortical neurons. We therefore aimed to establish a model system of peripheral motor neuropathy using human-induced pluripotent stem cell-derived neurons to investigate whether different ions, which have been shown to be released from implants, might have differential toxicity to human motoneurons.

## 2. Materials and Methods

### 2.1. Cell Lines

The human cell line used in this publication was published previously [20]. Fibroblasts originated from a healthy human female donor aged 48 at biopsy and were reprogrammed with standard Yamanaka factors (OCT4, KLF4, SOX2, and MYCC) delivered with the Sendai virus, as published [21]. Induced pluripotent stem cells (iPSC) were proliferated in mTESR1 (#85850, STEMCELL Technologies; Vancouver, BC, Canada) and characterized for pluripotency, trilineage differentiation, silencing of transgenes, and mycoplasm contamination [20]. The fully established iPSC lines were then converted to small molecule neuronal progenitor cells (smNPCs) using a previously published dual SMAD inhibition protocol [22].

### 2.2. Cell Culture

smNPCs were cultivated as previously published [23,24]. In short, cells were kept in N2SM1 (49% DMEM/F12 (21331020; Thermo Fisher Scientific; Waltham, MA, USA), 48.5% Neurobasal (21103049; Thermo Fisher Scientific; Waltham, MA, USA), 0.5% N2-supplement (17502048; Thermo Fisher Scientific; Waltham, MA, USA), 1% NeuroCult SM1 Neuronal Supplement (#05711; STEMCELL Technologies; Vancouver, BC, Canada), 1% penicillin–streptomycin–glutamine (10378016; Thermo Fisher Scientific; Waltham, MA, USA)) supplemented with 3 μM CHIR 99021 (2520691; Thermo Fisher Scientific; Waltham, MA, USA), 150 μM ascorbic acid (AA; A-4544; Sigma-Aldrich Chemie GmbH, Munich, Germany), and 0.5 μM purmorphamine (PMA; 10009634; Cayman Chemical; Ann Arbor, MI, USA). For differentiation, 1 million cells were seeded into 1 well of a 6-well plate and kept for two days in proliferation medium. Then, the medium was replaced with N2SM1 and supplemented with 1 μM retinoic acid, 200 μM AA, and 1 μM PMA and the cells were cultivated for 7 days. A quantity of 10^5^ cells was then seeded into 24-well plates and fed with N2SM1 supplemented with 2 ng/μL BDNF (AF-450-02; Thermo Fisher Scientific; Waltham, MA, USA), 2 ng/μL GDNF (AF-450-10; Thermo Fisher Scientific; Waltham, MA, USA), 1 ng/μL TGFβ3 (AF-100-36E; Thermo Fisher Scientific; Waltham, MA, USA), 200 μM AA, and 100 μM dbCAMP (1698950; Thermo Fisher Scientific; Waltham, MA, USA) until treatment. After, two weeks of maturation treatment with 0, 5, 50, and 500 μM CoCl_2_ (60818; Sigma-Aldrich Chemie GmbH, Munich, Germany), NiCl_2_ (451193; Sigma-Aldrich Chemie GmbH, Munich, Germany), or CrCl_3_ (27096; Sigma-Aldrich Chemie GmbH, Munich, Germany) was started. The media was replaced with supplemented metal ions every other day over the course of one week. All analyses were conducted after one week of treatment, resulting in a final maturation time of three weeks (Figure 1a).

### 2.3. NeuO Staining

NeuroFlour NeuO (#01801, STEMCELL Technologies; Vancouver, BC, Canada) was used to stain living cells that underwent our neuronal differentiation protocol. This allowed us to distinguish between neuronal and remaining non-neuronal cells. Staining was done according to the manufacturer’s instructions. In short, mature motoneurons were incubated with 0.5 μM NeuO in N2SM1 maturation medium for 1 h at 37 °C. The supernatant was aspirated and replaced with fresh media and the cells could recover for 2 h before analysis.

### 2.4. ROS Measurement

In order to investigate the abundance of reactive oxygen species (ROS), the cells were incubated 30 min prior to FACS analysis with 1 μM MitoSOX (M36007; Thermo Fisher Scientific; Waltham, MA, USA).

### 2.5. Fluorescence-Assisted Cell Sorting

For fluorescence-assisted cell sorting (FACS) analysis, cells were harvested by means of enzymatic detachment. Cells were incubated with accutase (A6964; Sigma-Aldrich Chemie GmbH, Munich, Germany) for 10 min at 37 °C. The reaction was then diluted with a 1:1 mixture of DMEM/F12:Neurobasal and the cell suspension was centrifuged for 5 min at 240× *g*. The cell pellet was then resuspended in FACS buffer (5% FBS (10437-028; Thermo Fisher Scientific; Waltham, MA, USA) in PBS (P04-36500; PAN-Biotech; Aidenbach; Germany)).

To assess the number of living cells, 16.66 μg/mL propidium iodide (421301; PerkinElmer; Waltham, MA, USA) were added to the sample containing FACS buffer.

The FACSCalibur (BD Biosciences; San José, CA, USA) was set up according to the manufacturer’s instructions. Detector voltages were adjusted one time at the beginning of the experiment and not changed during the measurement. In addition to the treated samples, there was always one unstained control sample (untreated, w/o MitoSOX, w/o PI) measured to check for background and auto fluorescence. For each condition, 50,000 cells were analyzed. For some experiments, not enough cells could be harvested due to cell death during the treatment time.

### 2.6. Statistics

FACS data of live/dead stainings were evaluated with a one-way ANOVA and Dunnett’s multiple comparison post hoc test vs. control condition.

For MitoSOX evaluation, raw fluorescence values were first logarithmized to obtain normal distributed data. From this normal distribution, the mean value was calculated and used to conduct the Kruskal–Wallis test over all biological replicates.

## 3. Results

### 3.1. Cell Death Caused by Metal Ions Is Dose Dependent

smNPCs were differentiated for 2 + 7 days in patterning media and 14 days in maturation media to derive mature neuronal cultures. The cells were subsequently incubated for 7 days with metal ions and a mock control and then processed for analysis (Figure 1a). When we treated the matured neurons with increasing doses of metal ions and subsequently analyzed them with FACS, we noticed that all of our samples yielded sufficient cells for analysis except two conditions. Neurons incubated with 500 μM Co^2+^ and 500 μM Ni^2+^ yielded less than 50,000 cells in total, despite 100,000 cells being seeded in all conditions (Figure 1b). In the case of Ni^2+^ this reduction was significant. This indicated that for the highest concentrations of Co^2+^ and Ni^2+^ there was tremendous cell death. Intrigued by this first finding, we proceeded to evaluate in detail which cell populations were most affected by the metal ion treatment.

**Figure 1 jfb-14-00392-f001:**
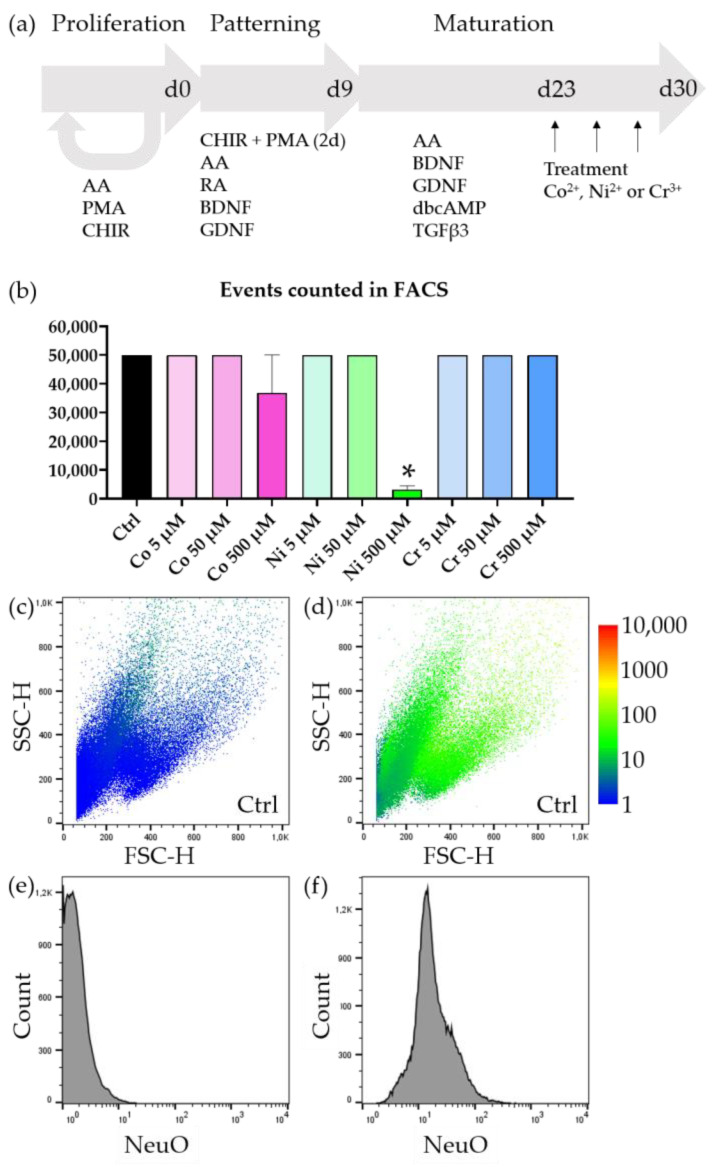
Detection of neurons by FACS. (**a**) Schematic of the iPSC-derived moto neuron differentiation protocol. (**b**) In each treatment condition 100,000 cells were plated. We aimed to analyze at least 50,000 cells per condition, which was not possible for 500 μM Co^2+^ and 500 μM Ni^2+^, which already indicated severe cell loss. (**c**) Representative FSC/SSC blot of an untreated and unstained sample—NeuO intensity colored (1—blue to 10,000—red). (**d**) Representative FSC/SSC blot of an untreated but stained (PI + NeuO) sample—NeuO intensity colored. (**e**) Histogram of an untreated and unstained sample. The FL-1 channel (ex: 480 nm; em: 530/30) represents the expected background signal for the NeuO staining. (**f**) Histogram of an untreated but stained (PI + NeuO) sample. There is a strong increase in green fluorescence compared to unstained samples, indicating successful incorporation of the dye. There are no observable individual populations, reflecting the neuronal nature of our culture system, in which only neuronal progenitor cells and terminally differentiated motoneurons exist. Bars represent mean ± SEM, *n* = 3, * represents *p* < 0.05.

### 3.2. Motoneurons Are Selectively Vulnerable to Metal Ions

Observing the FSC/SSC plot revealed 3–4 populations (Figure 1c,d). From our experience with the herein used differentiation protocol, we already knew that we could expect undifferentiated smNPCs and midbrain/hindbrain neurons enriched in motor neurons as the main cell populations [24,25]. In order to distinguish between differentiated neurons and undifferentiated progenitor cells, we used NeuO, a membrane permeable dye, which was reported to selectively stain living neurons. When we compared unstained, untreated differentiated neurons with stained, untreated neurons, we saw a strong increase in fluorescence (Figure 1e,f). However, this increase was observed in all events—except the low granular, small fraction—and we were not able to discern an independent population that exhibited an exceptionally strong NeuO signal (compare Figure 1c,d).

Continuing further with our analysis, we investigated PI incorporation. When overlaying the PI intensity in the FSC/SSC plot, we immediately observed that the previously identified populations could be characterized by their PI content (Figure 2a). The population characterized by larger, less granular features incorporated no PI and was therefore indeed a vital population, whereas the more granular population showed strong PI incorporation. When gating the PI channel and investigating the FSC/SSC plot, we identified two distinct populations (Figure 2b–d): one containing larger, less granular objects and one with very small objects. These small objects were very likely fragmented parts of dead cells that happened to not contain DNA and were therefore PI negative as well. We proceeded to gate the larger population and applied this double-gating strategy (PI-low, Living Neurons) to investigate the amounts of neurons in all conditions.

Treatment with metal ions showed an increase in cell death after one week (Figure 2e). This cell death was dose dependent, and higher concentrations of metal ions led to an increase in cell death—in particular, in the case of 500 μM Co^2+^ and 500 μM Ni^2+^ but not in 500 μM Cr^3+^.

### 3.3. Neurotoxicity Is Accompanied by Increase in ROS

In order to elucidate the cause of cell death, we speculated that ROS might be involved due to the nature of heavy metal ions. We applied MitoSOX to investigate levels of ROS with subsequent FACS analysis. When we observed the FSC/MitoSOX plot, we saw multiple populations. Under untreated conditions, there was a population of larger cells that exhibited low MitoSOX intensity and multiple populations consisting of smaller sized cells with varying MitoSOX intensities. These intensities ranged from being as low as the previous large cell population up to signal intensities stronger than two orders of magnitude (Figure 3a). Interestingly, in samples treated with 500 μM Co^2+^ or 500 μM Ni^2+^ there was an intermediate population with the same cell size but a more intense MitoSOX signal (Figure 3b–d). When comparing the different treatment conditions, it became evident that this intermediate population was only present in the samples treated with 500 μM Co^2+^ or 500 μM Ni^2+^. Since these samples also showed the strongest neurodegeneration, we speculate that this population was likely a transient stage and consisted of neurons that had already generated a lethal dose of ROS and were subsequently dying or already dead. The remaining living neurons showed little to no increase in ROS, which was very likely a survivor bias. When we gated the high and low MitoSOX populations individually and analyzed the FSC/SSC plot, we identified the low MitoSOX population occupying the same sector as the previously identified living neurons and the high mitoSOX population as the dead cells (Figure 3e,f). When we quantified the populations according to their ROS level, we were able to reproduce the reduced number of living neurons in the cells treated with 500 μM Co^2+^. This was accompanied by a significant increase in high MitoSOX cells. The intermediate populations we observed between low and high MitoSOX had a significantly increased MitoSOX signal compared to the neuron population, which was, however, still significantly lower than the high MitoSOX population’s signal intensity (Figure 3h). This intermediate population likely presented a transient stage in which cells had already accumulated a significant number of ROS and might not have been able to recover. Since we did not observe this population in lower concentrations, this might suggest that this population preceded the dead cells and that cells within this population were already beyond a point of no return and would inevitably have succumbed to apoptosis. 

## 4. Discussion

In this study, we demonstrate that our previously published cell culture model of human motoneurons from smNPCs derived from iPSCs can be used to model peripheral neuropathy due to environmental toxins [23,24]. The ions investigated in this study are used in alloys of contemporary metal joint replacements and can be detected in *in vivo* as well as in *in vitro* wear studies [6,7,8,26]. The concentrations used can be reasonably detected in patients and are well reported [13,14]. Furthermore, peripheral motoneurons are of particular interest, as they might be directly exposed to these ion loads since they are in direct contact with the affected peri-implant tissue. We show here that motoneurons died when exposed to metal ions. This effect depended on the ion type and concentration. Bivalent Co^2+^ and Ni^2+^ showed a stronger effect in our setup than trivalent Cr^3+^. This toxicity induced morphological changes by decreasing the cell size and increasing the granularity. This increase could potentially imply an abundant formation of stress granules or vacuolization. These hypotheses require further investigation. The cell death could be mediated by apoptosis. This programmed cell death can have different reasons. Due to the absence of lymphocytes, we can rule out T-cell-mediated cytotoxicity. Intrinsic apoptotic pathways follow environmental hazards like radiation, hypoxia, or other toxins and are mediated by changes in the mitochondria [27]. The observed phenotypes are very likely linked to ROS, as indicated by our analysis of MitoSOX. The presence of increased ROS upon metal ion treatment suggests that ferroptosis could play a major role in the neurotoxicity. This process is a distinct mechanism of cell death compared to other forms of apoptosis and is characterized by the accumulation of lipid peroxidation with an accompanied depletion of redox capacity [28]. The primary resolve mechanism to counter excessive lipid peroxidation is mediated by glutathione peroxidase 4 (GPX4) [29]. GPX4 can reduce toxic peroxides to alcohols by using glutathione. Since an increase in ROS challenges the cellular redox system, the lack of glutathione could lead to an increase in lipid peroxidation. Indeed, ferroptosis has been described in neuronal cells for cobalt nanoparticles as well as Co salts [30]. Furthermore, inhibiting ferroptosis induced by cobalt nanoparticles by alpha lipoic acid has been successfully shown [31]. An induction of ROS has also been shown in monocytes and macrophages upon Co and Ni treatment [32].

A major difference between the species of cobalt and nickel on the one hand and chromium on the other hand is their valance. Bivalent cobalt and nickel can activate the extracellular Ca^2+^ sensor receptor and trigger calcium-dependent pathways like p62 phosphorylation, whereas Cr^3+^ cannot interact in this way [33,34]. 

Metal ion-induced redox stress can be established by depletion of glutathione and protein-bound sulfhydryl groups [35]. This mode of action has been reported for ions like Cd, Ni, and Hg. Chromium, on the other hand, along with Fe, Cu, V, and Co, is known to act by redox cycling [35]. For chromium, a cycling through a tetra-, penta-, and hexavalent state has been reported. Interestingly, Cr^3+^, which was used in this study, is rather stable and does not directly challenge the redox system but rather tends to accumulate. One of the reported effects of Cr^3+^ is the inhibition of the mitochondrial thioredoxin reductase TrxR, which is pivotal for redox signaling, mitochondrial function, and cell survival [36]. Therefore, Cr^3+^ can indirectly challenge the redox system. Furthermore, the requirement of Cr^3^ for some cellular processes suggests that there might be a higher tolerance for this particular metal ion.

Titanium-based alloys like nitinol are often used because they are very wear and corrosion resistant [9]. Nickel, which can be released from these alloys, has been identified as causing ROS stress similar to in our study [37]. In the context of titanium, the predominantly studied species is TiO_2_, which is very stable and is linked to DNA damage and ROS [38]. Pure titanium and titanium alloys were not sufficient to induce changes or release of lactate dehydrogenase, a sign of cell death, in human fibroblasts [39]. Rat macrophages responded to treatment with Ti-Al-V by secretion of pro-inflammatory mediators, which caused secondary degeneration of the peri-implant tissue. However, the reaction was less severe than when treated with Co-Cr, which caused a highly toxic response [40].

The differentiation protocol that we used in this study yielded a high percentage of motoneurons [20,22,23]. Since motoneurons are in the close vicinity of skeletal muscles and joints, by using an MN specific protocol we hoped to mimic a realistic scenario. Other neurons, which should be similarly affected, are sensory neurons. Indeed, sensory neurons suffer from similar vulnerability to oxidative stress and can be affected by the wear and corrosive products of metal implants. Whether their vulnerability exceeds that of motoneurons needs to be investigated in further experiments. Because the second motoneuron and sensory neurons are peripheral nerves, the question arises of how cortical neurons are affected by systemic metal toxicity. Studies suggest that metal ions such as Co^2+^ can be readily found in cerebrospinal fluid and therefore must have crossed the blood–brain barrier [41,42]. Since the ion source for the systemic distribution is the implant itself, the concentration in the CNS will be lower than in the peri-implant tissue. Nevertheless, a rat model that uses direct cobalt chloride application to induce epilepsy has been described as well and indicated a direct effect of cobalt on the CNS [43]. A recent study also attributed an increased risk incidence of Parkinson’s disease or ischemic stroke to patients who have undergone orthopedic surgeries with iron-based implants [44]. In addition to ions, nanoparticles from implant wear can also be detected in patients and have been shown to promote inflammation *in vivo* as well [45]. Whether nanoparticles have an effect on motoneurons was not addressed by our study and needs further investigation. Although we did not investigate particular pathways, it is known that some metal ions like Co and Ni species can induce HIF1α and therefore may activate downstream stress pathways or lead to an decrease in heat shock factors that otherwise would be needed for neuronal homeostasis [17]. Since we demonstrate that there was an increase in ROS, it is plausible that downstream of metal ion intoxication mitochondrial homeostasis and oxidative stress, including mitochondrial and genomic DNA damage, appear. All of these factors are well known to be involved in neurodegeneration [46,47]. Hence, monitoring of peripheral (moto-)neuronal impairment and also signs of neurodegenerative diseases in general is important in patients undergoing total joint replacement, especially in those with expected long-term use of join replacements.

## Figures and Tables

**Figure 2 jfb-14-00392-f002:**
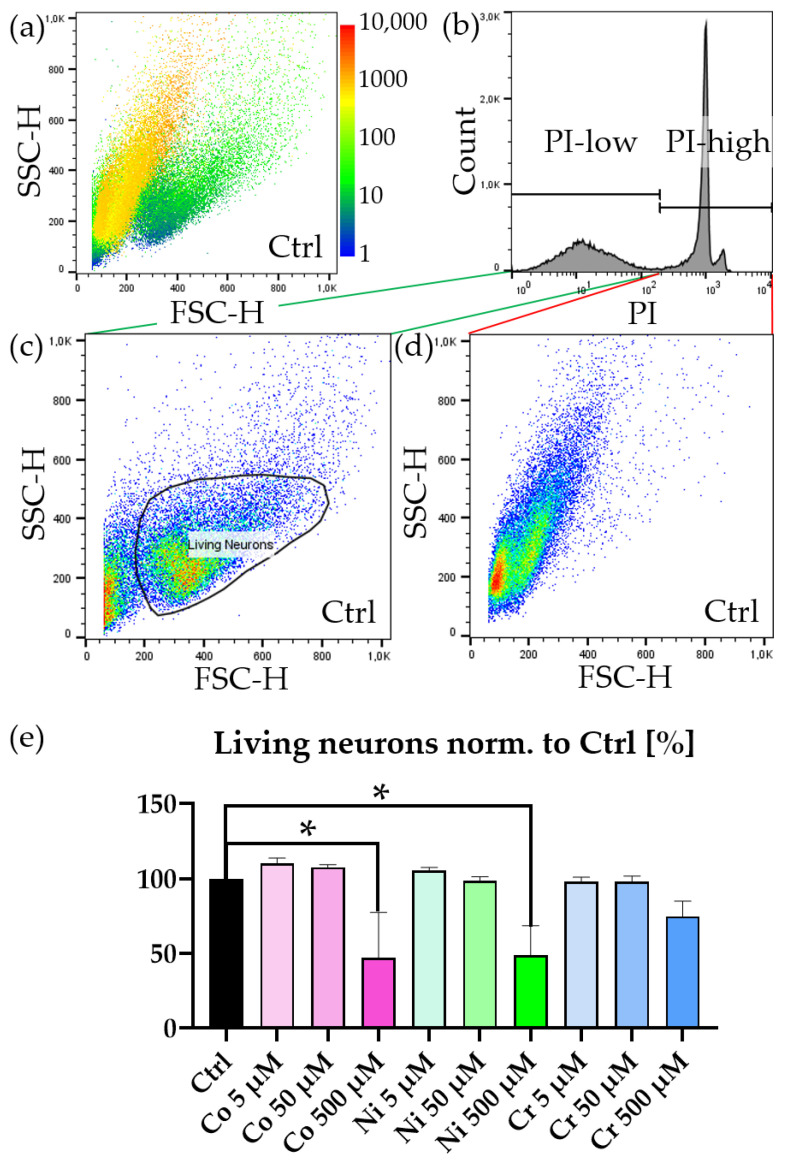
PI staining of metal ion treated neurons. (**a**) Representative FSC/SSC blot of an untreated but stained (PI, NeuO) sample—PI-intensity colored (1—blue to 10,000—red). (**b**) Histogram of an untreated but stained (PI, NeuO) sample. PI-low-gate contains unstained and therefore living cells with no PI incorporation, PI-high-gate contains stained and therefore dead cells with PI incorporation. (**c**) FSC/SSC blot of an untreated but stained (PI, NeuO) sample—pseudocolored (low density—blue to high density—red). Two populations are visible. One corresponds to living neurons (gate: “neurons”). The other population consists of cellular debris w/o DNA, which is not PI positive. (**d**) Cellular debris and dead cells, which were stained positive for PI. (**e**) Quantification of the population of living neurons (gating strategy: PI-low → neurons). Graph is normalized to untreated but stained condition “Ctrl.” Bars represent mean ± SEM, *n* = 3, * represents *p* < 0.05.

**Figure 3 jfb-14-00392-f003:**
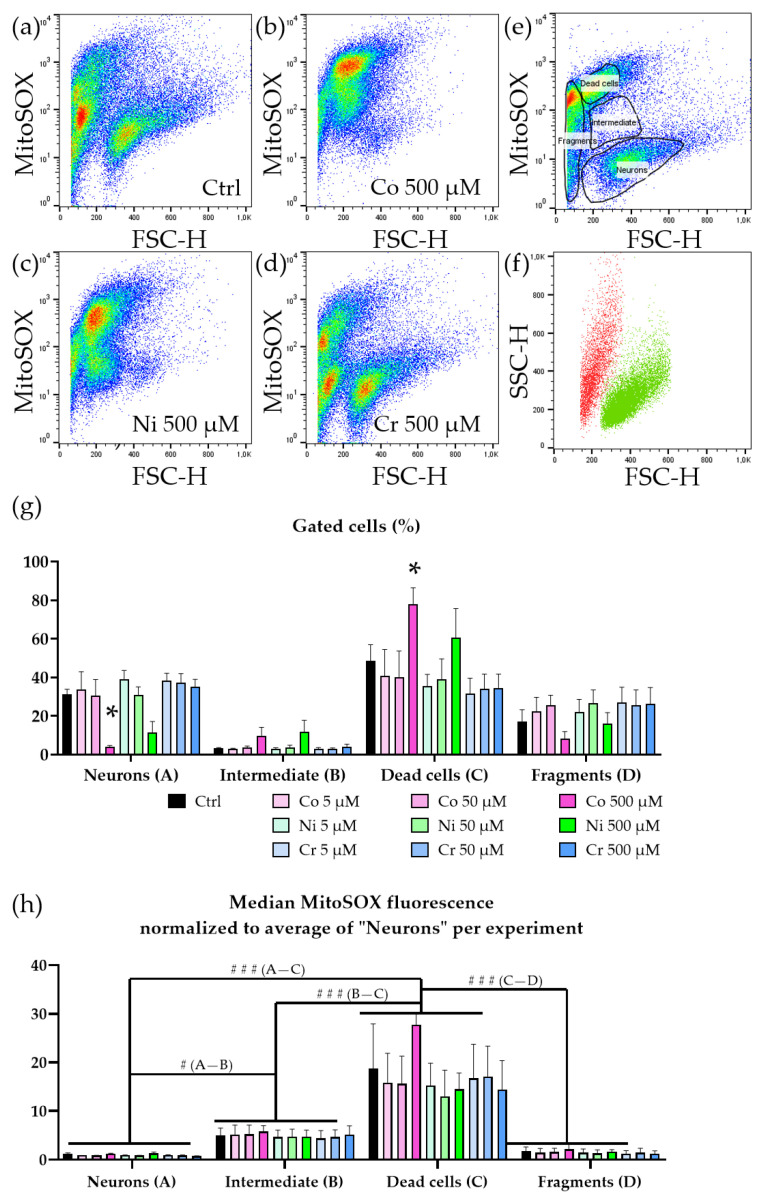
Increased ROS in metal ion-treated motoneurons. (**a**) Representative pseudocolored FSC/MitoSOX blot of untreated but stained (MitoSOX) sample. (**b**) Representative pseudocolored FSC/MitoSOX blot of treated (500 μM Co^2+^) and stained (MitoSOX) sample. (**c**) Representative pseudocolored FSC/MitoSOX blot of treated (500 μM Ni^2+^) and stained (MitoSOX) sample. (**d**) Representative pseudocolored FSC/MitoSOX blot of treated (500 μM Cr^3+^) and stained (MitoSOX) sample. (**e**) Representative gates for a population showing a high MitoSOX signal and a population showing a low MitoSOX signal. (**f**) Representative FSC/SSC blot of the gated populations “high MitoSOX” and “low MitoSOX”. The low MitoSOX population correlates with the previously described “living neurons” population and “high MitoSOX” is covering the area of dead cells. (**g**) Quantification gated populations “neurons”, “intermediate”, “dead cells”, and “fragmented”. The overall population of “intermediate” cells is very small compared to those of “neurons” and “dead cells”, indicating that this intermediate state might only be present during a short time period. (**h**) Quantification of median MitoSOX fluorescence normalized to the average of the “neurons” population within each experiment. Both “intermediate” and “dead cells” show a strong increase in MitoSOX signal, indicating high levels of ROS and oxidative stress. The “fragmented” population has a low MitoSOX level because cellular debris cannot retain the MitoSOX dye. Bars represent mean + SEM, *n* = 3, * represents *p* < 0.05 within a group, #/### represents *p* < 0.05/<0.001 between groups.

## Data Availability

The data that support the findings of this study are available on request from the corresponding author. The data are not publicly available due to privacy or ethical restrictions.

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
