# Peer review of "Corrosion Products from Metallic Implants Induce ROS and Cell Death in Human Motoneurons In Vitro"

_jfb, 2023, doi:10.3390/jfb14080392_

Round 1
Reviewer 1 Report
Dear Authors,
The brief report is really interesting, well conducted, and fits the objectives of the journal and special issue; but it is necessary to review some points in order to improve the quality of the paper:
Keywords should reflect the key points of the work and also they are used to assist search engines to find the article. Don't use abbreviations.
Abstract. There is no clear definition of what this work is about? what issues are being discussed in this brief report? what theoretically and practically significant results are obtained on the basis of this work? Therefore, the abstract requires changes and revisions in order to eliminate the above-mentioned shortcomings.
The ‘Nitinol’ alloy also should be added to the sentence “The most prominent metal alloys used in orthopedic surgeries are titanium-aluminum-vanadium as well as cobalt-chromium-molybdenum.”. Then, this sentence will be required proper references for each alloy: https://doi.org/10.1016/j.isci.2020.101745; https://doi.org/10.3390/ma13194292; https://doi.org/10.1016/j.cobme.2022.100429
The reference number should be in brackets ‘[]’.
The authors claimed that “The presence of increased ROS upon metal ion treatment suggests that ferroptosis could play a major role in the neurodegeneration.”. Please provide more explanation and discussion around this matter.
Authors should pay attention to the technical design of the work and the elimination of typos and errors, for example in the section “Neurodegeneration is accompanied by increase in ROS”.
Minor editing of English language required!
Reviewer 2 Report
The authors have presented an interesting topic. The manuscript is well written . The results were comprehended nicely and correlated well. However, I note the following?
1. Why ROS study was restricted to only Co and Ni? There are others metallic ions like Cr, Ti , authors did not mention anything about that.
2. Concentrations used in the study were based on literature. However, did authors try evaluating any range of concentations (specially lower and upper limit of the concentations) which could provide an useful insight
3. Relation of ROS with neurodegeneration requires more clarity.
Reviewer 3 Report
The current work by Glaß et al, used human iPSC derived motoneurons as a model of peripheral neuropathy to investigate the effects of metal ions on neurons. The study should be revised to improve several aspects, as described below:
1. As the study performed only FACS for detection of neurons and ROS, the authors should rephrase the title of the manuscript which is more specific to the work they have done.
2. The authors in the last sentence of the abstract claim that the ‘model can be used to model secondary neurodegenerative diseases.’ However, more investigations are required in the current work before this claim can be made.
3. More findings from the paper could be included in the abstract while removing the redundant information.
4. Include a discussion into the type of neurons that might be affected due to these metal ions and the type you have used to test these.
5. Include in discussion a paragraph about the possible mechanism of action for these metal ions.
6. Provide some supporting results for the established cell line (smNPCs) or provide a reference for the successful establishment of cell line. Also, specify the source of cells (human/animal)?
7. Wherever authors have mentioned ‘previously published’, they should provide appropriate reference for the same.
8. During the treatment phase were the CoCl2, NiCl2, CrCl3 supplemented? Provide details in the manuscript.
9. For the concentration of metal ions used how was the concentration selected? There seems to be a large gap in concentration range from 50 to 500 μM. There might be a critical value in between this larger range. Explain the reason for this in the manuscript.
10. Expand FACS as Fluorescence assisted cell sorting (line 112).
11. Fig 1a – says analysis was performed 21 days after maturation, but the materials and methods sections describe differently. Please confirm this and make it less confusing.
12. All the figures need to be rearranged for better clarity and readability.
13. Fig 2H – statistic bars are confusing. Fig 2 g-h – the histograms are too small in width.
14. Fig 2 – provide more explanation of the obtained results. For example, what is Intermediate cell population and why it might have been generated.
15. Line 224: ‘Cr3+ seems to act upon a different pathway’. Explain this further and hypothesize a reason for this supported with literature.
There are many grammatical mistakes and typos in the manuscript. The whole manuscript needs to be checked for grammar and typo errors. For example, Line 20 (extra s), line 39 (correct the sentence for grammar), Line 95 (TGFb3), line 217 (extra n after group), etc.
Round 2
Reviewer 1 Report
The authors justified the queries made by this reviewer. I think the revised version may be accepted in its current form.
Minor editing of English language required!
Reviewer 3 Report
The manuscript may now be accepted.